# Detection of the New Class of Hypersonic Targets under Emerging Hyperspectral Sample Streams: An Unsupervised Isolation Forest Solution

Shurong Yuan [1,2,†], Lei Shi [1,2,*,†], Bo Yao [1,2], Yutong Zhai [1,2], Fangyan Li [1,2] and Yuefan Du [1,2]

[1] Key Laboratory of Equipment Efficiency in Extreme Environment, Ministry of Education, Xidian University, Xi'an 710071, China
[2] School of Aerospace Science and Technology, Xidian University, Xi'an 710071, China
* Correspondence: lshi@xidian.edu.cn; Tel.: +86-029-81891052
† These authors contributed equally to this work.

**Abstract:** Rapid detection of the new class of hypersonic targets (HTs) presenting unknown military threats in space-based surveillance will guarantee aerospace security. This paper proposes an unsupervised subclass definition and an efficient isolation forest based on an anomalous hyperspectral feature selection (USD-EiForest) algorithm to detect the new class of never-before-seen HTs under emerging hyperspectral sample streams. First, we reveal that the hyperspectral features (HFs) of the new class of HTs have no anomaly characteristics when compared to the globally observed samples while having prominent anomaly characteristics when compared to the subclasses of observed samples. Second, an unsupervised subclass definition method adapted to HTs is utilized to classify the observed samples into several subclasses. Then, an efficient isolation forest is designed to determine whether the data stream sample in each subclass indicates anomaly features that mark the detection of the new class of hypersonic targets (DNHT). Finally, we experiment on the simulated hyperspectral HTs data sets considering the RAM-C II HT as the observed samples and the HTV-2 HT as the unknown samples. The results suggest that the performance of our proposal has competitive advantages in terms of accuracy and detection efficiency.

**Keywords:** hypersonic targets; hyperspectral feature; unsupervised subclass definition; anomaly detection; density peak clustering

---

## 1. Introduction

Hypersonic targets (HTs), achieving significant space access and prompt global striking ability, have attracted much attention over the past few decades [1]. The classification, recognition and detection of the observed HTs are of great significance in aerospace security defense [2]. Their observed features are degraded owing to the high moving speed of the HTs, leading to difficulty in detection and recognition with radar and infrared detectors [3]. Fortunately, the HT surface will generate high temperatures from intense friction with the surrounding atmosphere during the flight state. It ionizes air molecules around the vehicle, developing a certain thickness of "plasma sheath" [4–8]. The generated plasma sheath usually shows a thermodynamic nonequilibrium state and will continuously emit strong spectral radiation, which can be used as a fingerprint feature to realize the detection and classification of HTs [9–13].

When using the hyperspectral features (HFs) of HTs obtained by space-based detectors for actual observations, greater emphasis should be placed on the unknown military threats of the new class of HTs that the detectors have not observed [14]. An algorithm should be proposed to distinguish whether the detected samples belong to previously observed classes or a newly emerged class [15]. Developing an annotated data set is difficult even if a certain number of HFs of the HTs observed with space-based hyperspectral detectors

is accumulated to determine the unique value of the HTs [16]. Therefore, we propose the detection of the new class of hypersonic targets (DNHT) method to be completed in an unsupervised manner.

Many industrial fields are also facing the problem of new class target detection. Some methods have been formulated for the detection of the new class as a two-class recognition problem (unknown versus known) where both positive and negative samples are relevant [17–19]. The common approach for detecting the new class is to collect both real and fake data and try to learn a suitable two-class classifier, employing a supervised or semi-supervised approach [20,21]. However, the collection of new HTs is difficult, and the detection methods based on two classes are not suitable for DNHT. There are also novelty detection methods that mainly identify data with unsupervised methods. Liu et al. proposed iForest [22], and Yang studied a KNN-based approach for unsupervised novelty detection [23,24]. Rettig et al. proposed online anomaly detection over big data streams [25], and the unsupervised real-time anomaly detection for streaming data was subsequently developed [26]. However, unsupervised new class detection requires a large difference between the features of the new class and the existing class samples, that is, anomaly characteristics. The interclass variability of the HFs of HTs might be smaller than the intraclass variability, which results in a high error rate for the new class detection method based on anomaly detection. Another line of work focuses on open-set recognition (OSR) [27], which aims to classify known classes and reject novel ones. OpenMax utilized Weibull-based calibration to augment the softmax layer and detect novel classes [28]. Proser proposed reserving the probability for novel classes during close-set training and transformed closed-set training into open-set training [29]. CPL optimized the embedding with margin-based classification loss for better feature extraction [30]. However, these OSR models must be trained with a large data set, and annotated observed HT data sets are currently unavailable. Additionally, the imbalanced hypersonic observations often degrade the network performance [31].

In general, the above algorithms will encounter two challenges when applied to DNHT. First, the accumulated observation samples are not labeled, and the new class HTs are difficult to obtain, which causes the learning of the two-class classifier and OSR framework to fail. Second, detecting the new class based on anomaly detection requires a large feature difference between the new class and previous observations. However, the intraclass variability of HTs is greater than the interclass difference, and the anomaly characteristic of the new class is not apparent compared with globally observed samples. To detect the new class hypersonic vehicle under streaming emerging HTs, we propose using the unsupervised subclass definition and the efficient isolation forest based on anomalous hyperspectral feature selection (USD-EiForest) method for DNHT based on unsupervised classification and anomaly detection, which has the following distinguishing features:

- Existing algorithms train classifiers in a supervised or semi-supervised manner to achieve new class detection. In contrast, the proposed USD-EiForest method employs an unsupervised learning model to solve the problem of DNHT in the case where the HFs data set of the HTs has not yet been constructed.
- The phenomenon is revealed, whereby the HFs of new class HTs have prominent anomaly characteristics relative to the local subclasses of observed samples. To take full advantage of the local anomaly characteristic for new class detection, we use the unsupervised subclass definition method based on density peak clustering (DPC) to achieve subclass division.
- Because the features of the observed targets are more concentrated and more conducive to anomaly detection, an efficient isolation forest algorithm based on anomalous HFs selection is proposed with high detection efficiency and accuracy for new class detection with respect to subclass samples.

The rest of this paper is organized as follows. Section 2 describes the intuition of the proposed algorithm. Section 3 describes the related definitions and details of the proposed

algorithm. We report the experimental results in Section 4, which show that the proposed method can accurately detect the new types of HTs. The conclusion is provided in Section 5.

## 2. Interclass and Intraclass Variation of HFs of HTs

The plasma sheath (thermochemical nonequilibrium flow field) is generated while an HT is flying in near space [2], which usually exhibits the stated thermodynamic nonequilibrium [32]. Research has revealed that the gas in the plasma sheath of the HT ($O_2$, $N_2$, NO, N, O, $NO^+$, $N_2^+$, $O_2^+$, $N^+$, $O^+$ and $e^-$) has a strong spectral radiation effect [33,34]. Although the HFs of HTs based on actual space-based detectors are unavailable for the development of HTs and are not comprehensive enough, acquiring HT samples will be straightforward when HTs are put into practical applications in the future. To alleviate the spatial redundancy of the spectral characteristics of the object, this section provides the spectral radiation characteristics at the place with the strongest spectral intensity as fingerprint features for classification and detection based on the numerical calculation results of the surface flow field of the RAM C-II and HTV-2 re-entry HTs. This provides data support for subsequent proposal verification and reveals the distribution of intraclass and interclass HFs of HTs.

The process of obtaining the spectral radiation characteristics of the HTs using the space-based spectral detector is shown in Figure 1. According to various gas kinetic models, the air plasma sheath can contain several different compositions, such as 5, 7 and 11 species. To consider the computational complexity and simulation accuracy of spectral radiation characteristics at the same time, this paper adopted the gas model of seven species ($N_2$, $O_2$, N, O, NO, $NO^+$, $e^-$). First, the emission and absorption coefficients that describe the radiative properties are of great importance to the calculations of radiative transfer in the thermochemical nonequilibrium flow field, which has been solved in the literature [13]. Second, the spectral radiation intensity on the surface of the plasma sheath that envelopes the HT was calculated with the line of sight (LOS) method, assuming that each layered medium is isotropic and isothermal; this was used to compute the radiation transfer equation with known emission and absorption coefficients [35]. Finally, the transmission model was established based on MODTRAN5 software to reflect the atmospheric attenuation effect of spectral transmission [36]. The brief process of the software is depicted in Figure 2, including atmospheric data input, aerosol data input, geometric data input and spectral information input. The 1976 American standard atmospheric model [37] was used to simulate atmospheric transmission. The spectral radiation characteristics of HTs picked up by the space-based detector after atmospheric attenuation were used for unsupervised classification. The details of the whole process can be found in Ref [38]. However, owing to the scarcity of plasma sheath flow field data of hypersonic targets, this paper only simulated the HFs of RAM-C and HTV 2 hypersonic targets under different typical flight states.

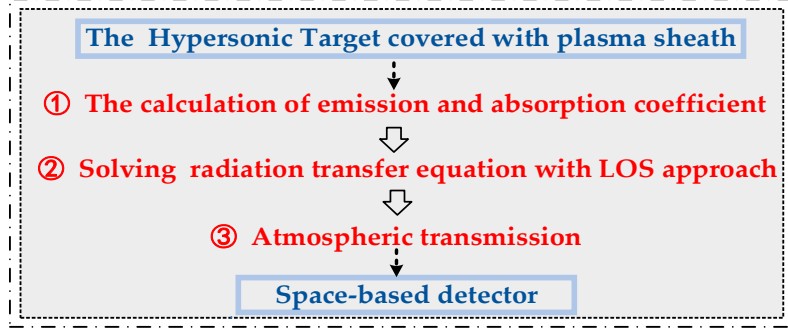

**Figure 1.** The flow chart for calculating spectral radiation characteristics for HTs.

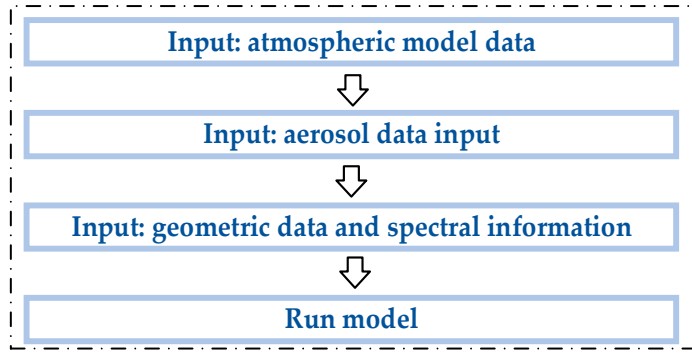

**Figure 2.** The brief MODTRAN5 process.

The spectral radiation characteristics of RAM-C II and HTV-2 HTs observed by space-based low earth orbit (LEO) detectors under different flight states were calculated based on the geometric model and the flow field distribution of these two HTs [38,39]. The results are shown in Figure 3, indicating the intensities of the calculated spectral features not of the same order of magnitude. Moreover, it is impracticable for space-based detectors to obtain all the spectral bands of radiation characteristics of HTs. Therefore, 120 wavebands were selected and normalized at the radiation intensity peak, as shown in Figure 4. To more clearly reflect the intraclass and interclass variation of the HFs of HTs, principal component analysis (PCA) was carried out on HFs, and the result is shown in Figure 5 and Table 1. The pink represents the HTV-2 HTs, and the blue represents the RAM-C II HTs. We can see in Figure 5 that the HFs of the same type of HTs were different when the flight states changed. When combined with the specific values in Table 1, this shows that different types of HTs flying in the same state have small HF differences.

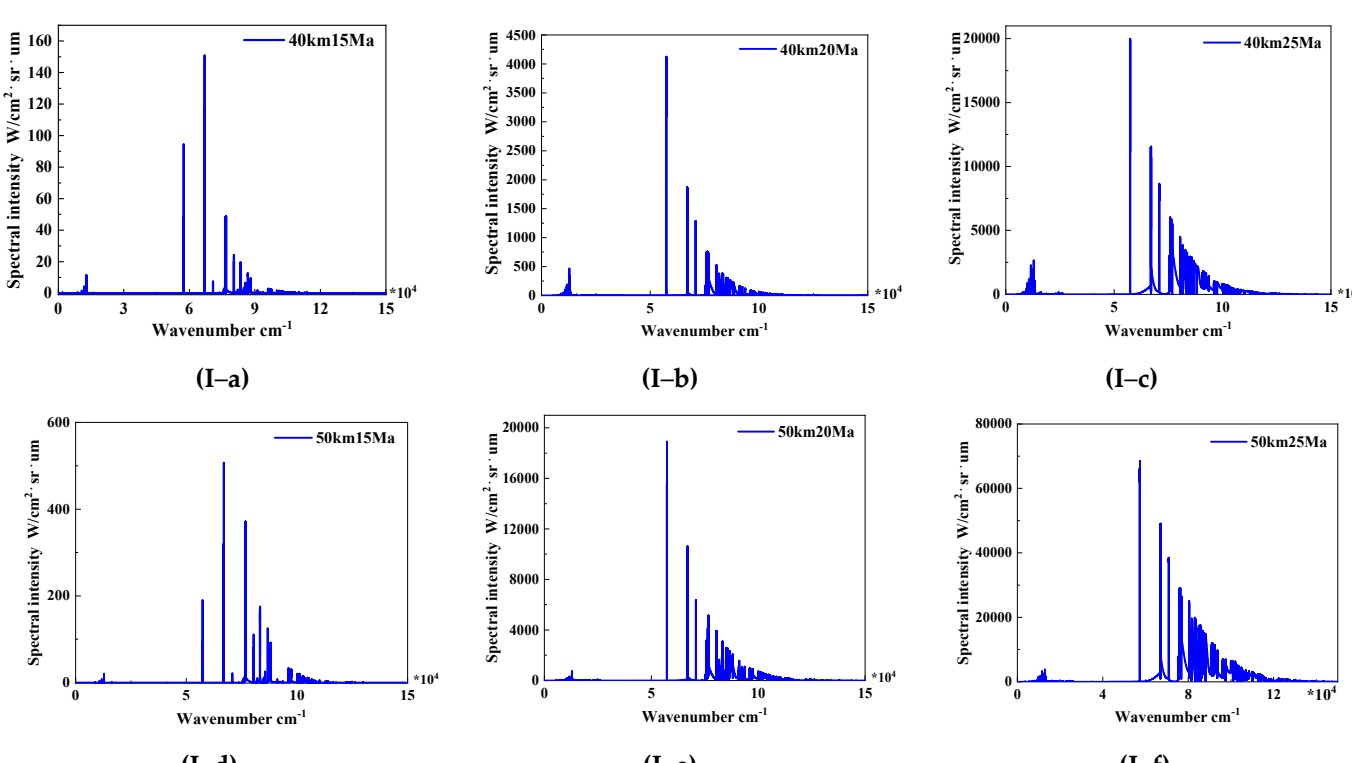

**Figure 3.** *Cont.*

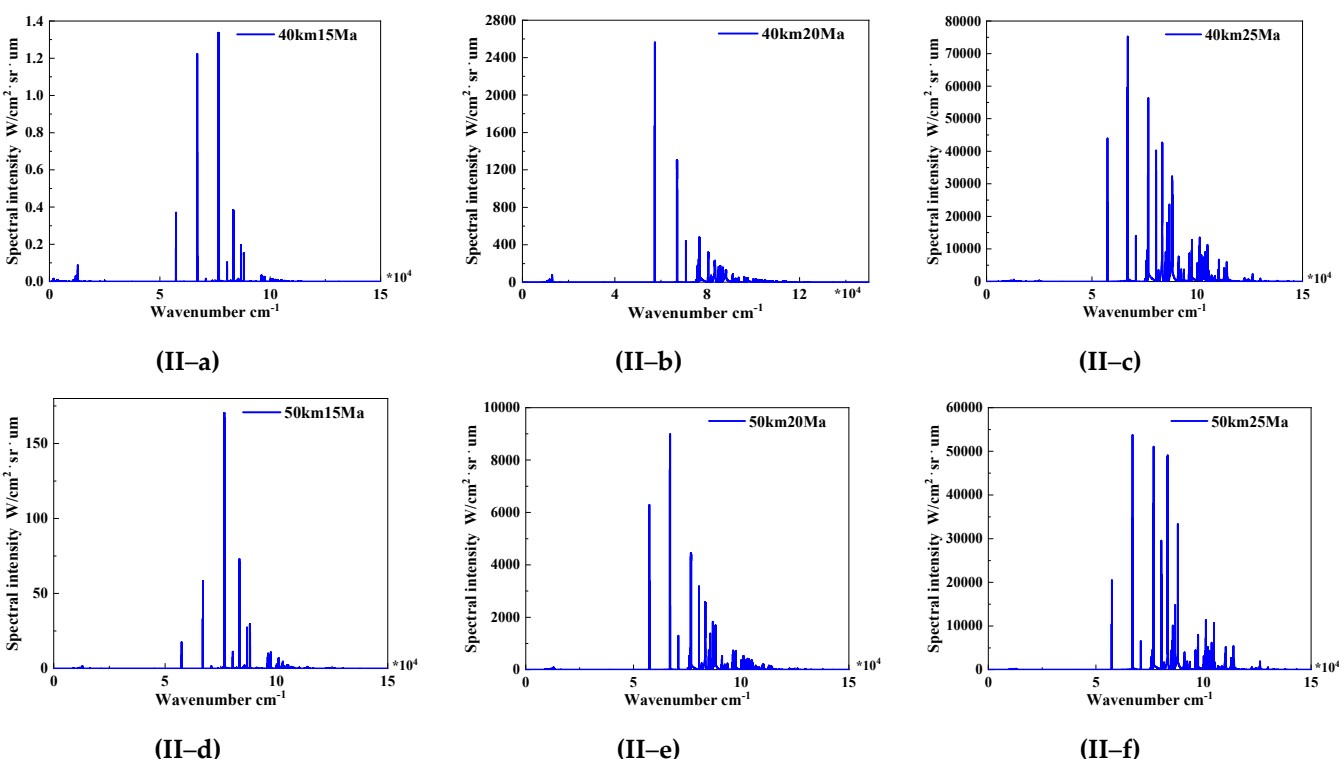

**Figure 3.** The simulated spectral radiation characteristic of HTs acquired by space-based detectors. (**a**) 40 km 15 Mach. (**b**) 40 km 20 Mach. (**c**) 40 km 25 Mach. (**d**) 50 km 15 Mach. (**e**) 50 km 20 Mach. (**f**) 50 km 25 Mach. (**I**) RAM-C II. (**II**) HTV-2.

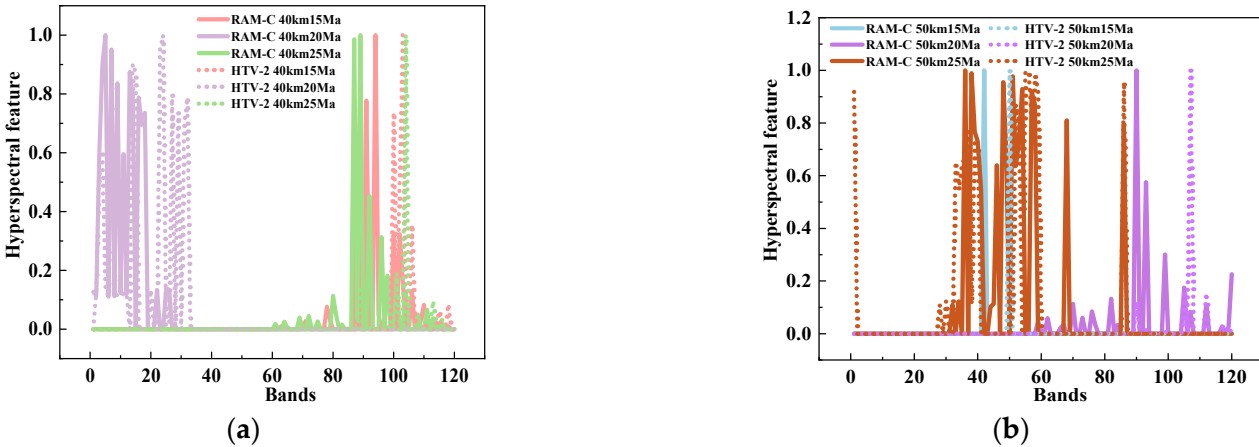

**Figure 4.** The HFs of RAM-C II and HTV-2 at typical flight states. (**a**) The flight states are 40 km 15 Mach, 40 km 20 Mach and 40 km 25 Mach. (**b**) The flight states are 50 km 15 Mach, 50 km 20 Mach and 50 km 25 Mach.

Anomaly detection techniques can be successfully used in the area of new class detection if it is understood that the instance of any developing new classes is distant from the known classes or at the edge of the data cloud of known classes. When RAM-C II was viewed as the known class and HTV-2 as novel class, it is evident from Figures 4 and 5 that the spectral radiation characteristics of HTV-2 did not differ significantly from those of the global RAM-C II samples. The interclass variability between the two classes of HTs was far lower than the intraclass variability, making it difficult to detect the new class using anomaly detection methods.

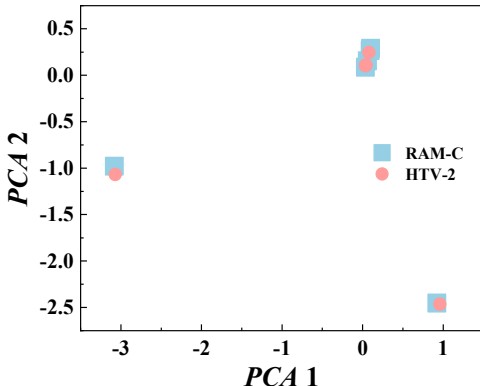

**Figure 5.** Extracted PCA features.

**Table 1.** The value of extracted PCA features.

| Flight States | RAM-C II | | HTV-2 | |
| --- | --- | --- | --- | --- |
| | PCA 1 | PCA 2 | PCA 1 | PCA 2 |
| 40 km 15 Ma | 0.09377 | 0.26556 | 0.07957 | 0.24599 |
| 40 km 20 Ma | 0.92226 | −2.45232 | 0.95889 | −2.46476 |
| 40 km 25 Ma | 0.10042 | 0.28921 | 0.03923 | 0.11015 |
| 50 km 15 Ma | 0.03547 | 0.08771 | 0.03653 | 0.10195 |
| 50 km 20 Ma | 0.05973 | 0.15594 | 0.03927 | 0.11026 |
| 50 km 25 Ma | −3.08021 | −0.97891 | −3.0709 | −1.06767 |

## 3. The USD-EiForest

### 3.1. The Core Idea of USD-EiForest

The regional distribution of abnormal and normal target features is depicted in Figure 6. Anomaly targets are usually found far away from the normal region or spread on the edge of the normal region. It can be observed in Figure 5 that when the region where the HFs of RAM-C II are located was regarded as the normal region, the HFs of HTV-2 belonged to the normal region and did not have an anomalous characteristic.

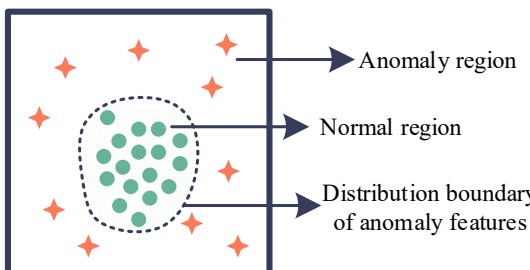

**Figure 6.** The description of anomaly feature distribution.

The new class of HTs had no global anomaly compared to the observation samples. Still, it had local anomaly features related to a subclass of observation HTs, which was the fundamental premise for proposing the USD-EiForest. The pink dots in Figure 7 reflect the newly emerging class HTV-2 with a flight state of 40 km 15 Mach. The blue point is the mean value of the HFs of the previously observed target at typical flight states after PCA dimensionality reduction. The blue area is the variation range of the HFs of the same observation class at a given typical flight state, where $r_i$ is the maximum variation range, and $i = [1, 2, 3, 4, 5, 6]$ is the $i$-th flight state. If $r_i$ is less than the difference of HFs between an observed target and a newly emerged class target, the new sample will present local anomalous characteristics compared with observed HTs under a typical flight state.

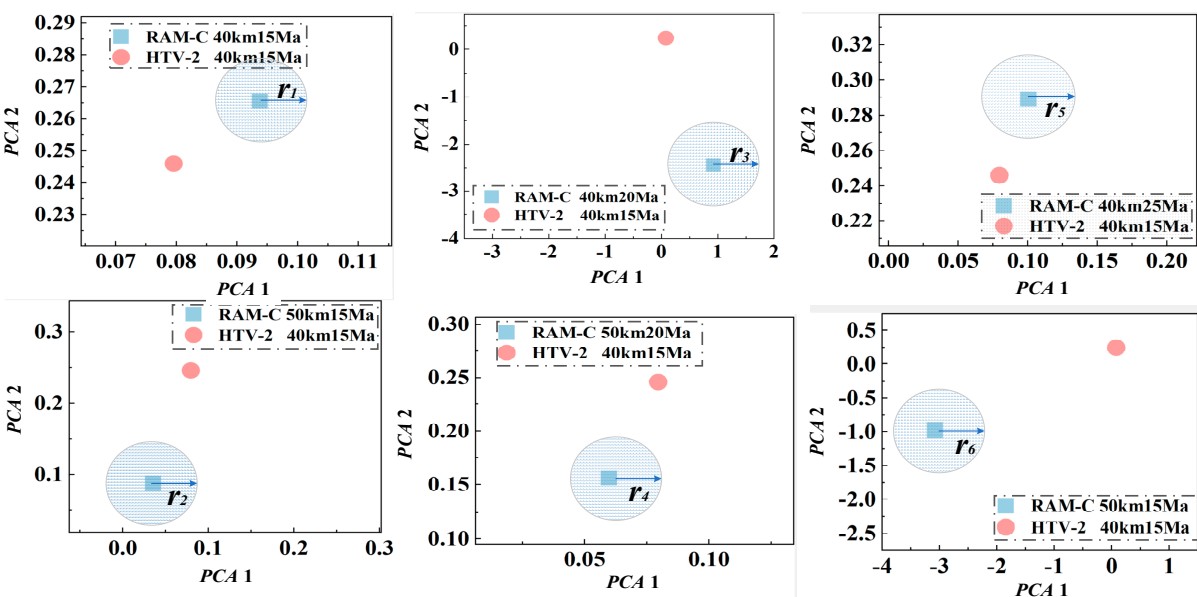

**Figure 7.** Local anomalies of a new class of HTs.

Based on the distribution of the HFs of HTs revealed above, we proposed the USD-EiForest method for DNHT. As shown in Figure 8, the proposed new class detection algorithm for HTs consisted of two steps. First, an unsupervised subclass definition (USD) algorithm based on DPC, dividing the existing accumulated samples into multiple subclasses, classified different flight states of the same HTs. Second, the improved efficient iForest (EiForest) was used to determine whether the emerging sample was an anomaly target compared to subclasses. If the emerging sample was an anomaly in each subclass, we concluded that it was a new class. Otherwise, it was a known sample. Both the unsupervised classification and the improved iForest are detailed in Section 3.2.

### 3.2. The Unsupervised Subclass Definition and Efficient Isolation Forest

The details of the proposed algorithm are described in this section, comprising unsupervised classification of subclasses and improved iForest. Each sample in a data stream is assigned a class label: emerging new class or one of the known classes. The pertinent details in the procedure are provided in the following sections.

#### 3.2.1. Unsupervised Subclass Definition

For the sample feature of the new class to conform to an anomaly characteristic instead of the normal characteristic, an unsupervised classification method was used to divide the observed samples into multiple subclasses. The overall flow chart is shown in Figure 9. The same color dots in Figure 9 constitute the same super node.

##### Intraclass Variation Mitigation

The difference in the observation angle, position, atmospheric environment, the complex internal structure of the detector and the spectral radiation mechanism of the hypersonic targets lead to the phenomenon of intraclass variability of the spectrum. This seriously affects the performance of the unsupervised classification of targets based on HFs. The HFs of the super nodes are jointly represented by median filtering on sub-nodes in themselves, which alleviate the phenomenon of intraclass variability of HFs. The observed samples are represented as $Y_{obs} = [Y_1, Y_2, Y_3, \dots, Y_m]$, where $m$ is the number of observed samples, and $Y_i = [y_{i1}, y_{i2}, y_{i3}, \dots, y_{if}]$. The observed samples after correction are shown as $Y_{obs}' = [Y_1', Y_2', Y_3', \dots, Y_m']$, where $Y_i'$ is the corrected HFs of the $i$-th observed sample, and $Y_i' = [y_{i1'}, y_{i2'}, y_{i3'}, \dots, y_{if'}]$, where $f$ is the number of HFs.

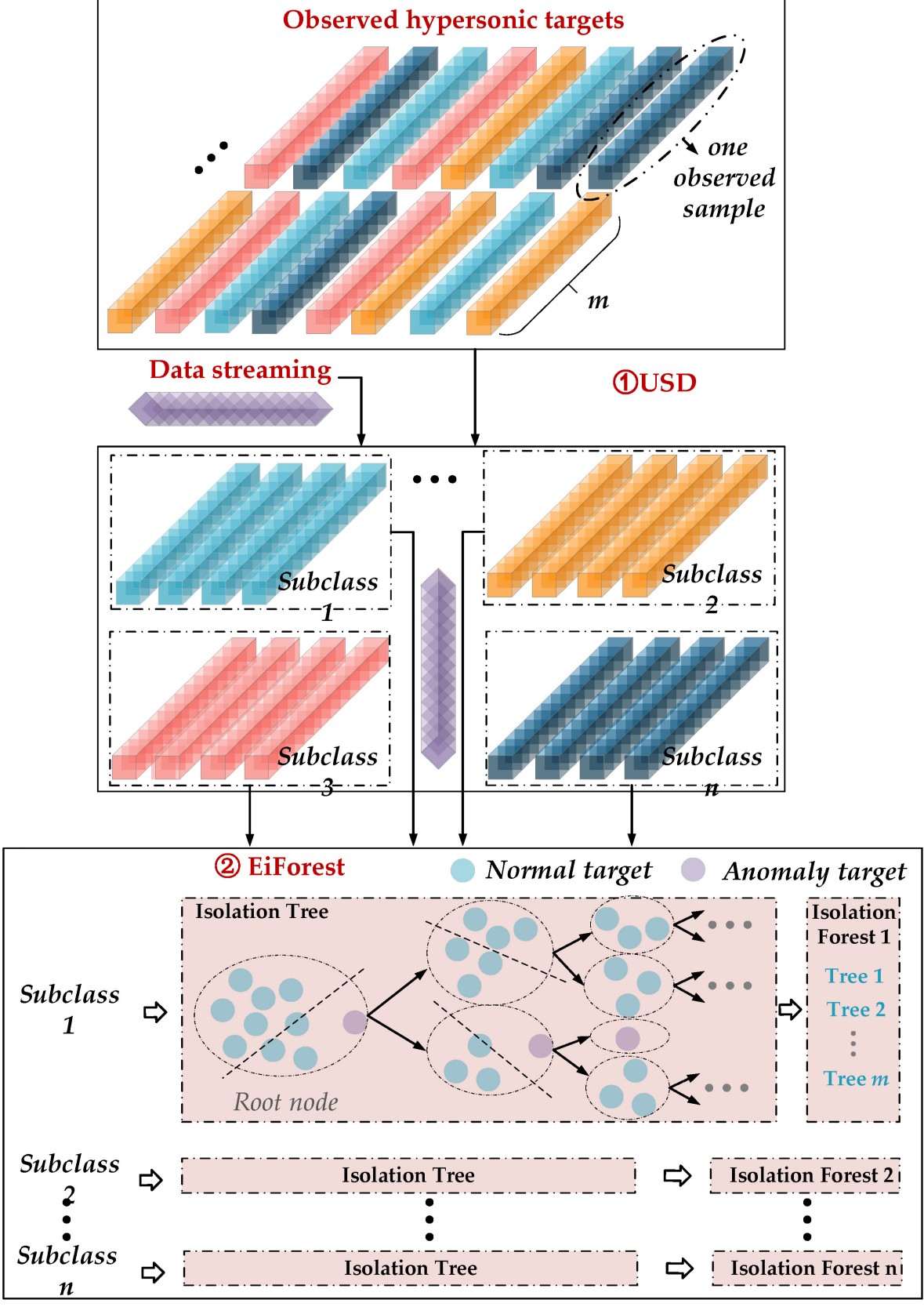

**Figure 8.** The proposed USD-EiForest method for DNHT.

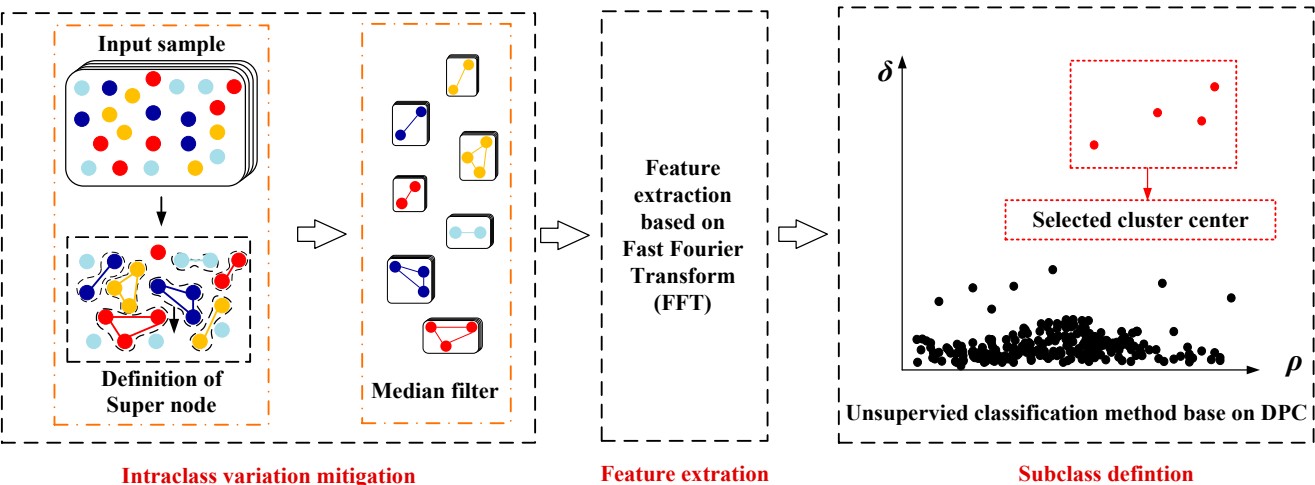

**Figure 9.** The flow chart of the proposed unsupervised subclass definition method.

Feature Extraction

The information on the fast and slow fluctuation of the HFs curve is extracted as extraction features. The low-frequency part of the spectrum curve represents the slow fluctuation of the spectral radiation characteristics of hypersonic targets, and the high frequency represents abrupt fluctuation, which can be extracted by fast Fourier transform (*FFT*) to distinguish the targets. When performing the $f$-point FFT operation on the HF of each super node, the first $k/2$ low-frequency and last $k/2$ high-frequency information are selected as the extracted features to implement dimension reduction from $f$ to $d$. The HFs of the $i$-th observed sample $Y_i'$ are represented as $X_i = [x_{i1}, x_{i2}, x_{i3}, \ldots, x_{id}]$ after feature extraction, where $d$ is the number of extraction features.

Subclass Definition

An unsupervised classification algorithm based on cluster peak selection [40] is used to find cluster centers and realize unsupervised classification without iteration. This step divides the observed HT samples into $n$ subclasses $[Y_{sub\_1}, Y_{sub\_2}, Y_{sub\_3}, \ldots, Y_{sub\_n}]$, and the center samples of subclasses will be saved as $[C_1, C_2, C_3, \ldots, C_n]$. The $j$-th subclass is represented as $Y_{sub\_j} = [Y_{j1}, Y_{j2}, Y_{j3}, \ldots, Y_{jeach\_j}]$, where *each_j* is the number of HT samples in the $j$-th subclass. The pseudo-code of the proposed subclass definition method for hypersonic targets is reflected in Algorithm 1.

---

**Algorithm 1**. The unsupervised classification algorithm.

---

**Input:** the spectral feature of hypersonic targets $Y_{obs}$, the number of vertex groups $N_{grounp}$.
**Output:** classification label for each sample.
1 Randomly divide the input $Y_{obs}$ samples into $N_{group}$ groups:
2 Calculate the adjacency matrix for each sub-group $V_{group\_i}$.
3 Define super nodes for each group $V_{group\_i}$ and gather all:
$\{SN_1, SN_2, \ldots, SN_q\}$ ($q$ is the number of total super nodes);
4 Calculate the spectral radiation feature of super node $SN_i$ based on median filter:
$Y_{obs}' \in 1 \times f$;
5 Feature extraction based on FFT for $Y_{obs}'$:
$E_i \in 1 \times d$;
6 Calculate the distance and density for $E_i$ based on DPC:
density $\rho_i$, and distance $\delta_i$;
7 Determine the cluster center;
8 Complete subclass definition according to cluster center.

---

### 3.2.2. The Efficient iForest

When performing new class detection for HT data stream samples, the high moving speed characteristic requires high detection efficiency. Isolation forest (iForest) [41], an anomaly detection algorithm, utilizes no distance or density measure and has a linear time complexity with a low constant and a low memory requirement, which was improved and applied to the recognition of a new class of emerging hypersonic samples in this paper.

Anomalous HFs Selection

Even though the anomaly detection based on iForest is efficient, the isolation trees (iTree) that comprise the iForest are constructed using random feature selection, which will bring additional computational redundancy in the anomaly detection of HTs.

Figure 3 shows that a specific HT sample is represented by 120 spectral bands, and only a few bands have obvious HFs. To improve the efficiency of iForest construction and remove redundant bands with weak spectral radiation, we set the spectral threshold $s_{th}$ at 0.02 and input a set of $n$ subclasses $[Y_{sub\_1}, Y_{sub\_2}, Y_{sub\_3}, \ldots, Y_{sub\_n}]$. The HF of each band in samples of the $j$-th subclass is compared with $s_{th}$, and the feature bands whose values are higher than $s_{th}$ are recorded into the set $[B\_j1, B\_j2, B\_j3, \ldots, B\_jeach\_j]$, where *each_j* is the number of hypersonic samples in the $j$-th subclass, and $B_{j1}$ is the selected bands from the first observed hypersonic samples of subclass $j$. Then, the feature band selected from subclass $j$ can be expressed as

$$B\_allj = B\_j1 \cup B\_j2 \cup B\_j3 \cup \ldots \cup B\_jeach\_j \tag{1}$$

where U means the union operation of the selected band. Figure 5 shows that the normal region is constituted by the features of observed samples. It is obvious that when the features of the observed targets are more concentrated, they are more conducive to anomaly detection. Therefore, the HFs with small variance are selected to remove the redundancy of bands, and the $j$-th subclass $Y_{sub\_j} \in R^{each\_j \times B\_allj}$ is input after removing the weak features. The variance of each band is expressed as

$$V\_j = [v_{j1}, v_{j2}, v_{j3}, \ldots, v_{B\_all}] \tag{2}$$

where the variance of the $m$-th band is expressed as

$$v_{jm} = \frac{\sum_{l=1}^{each\_j} (dist(y_{lm}, u_m))^2}{Card(Y_{sub\_j})} \tag{3}$$

where *Card* $(Y_{sub\_j})$ is the cardinality of $Y_{sub\_j}$, $u_m$ is the mean value of the $m$-th band of all samples in subclass $Y_{sub\_j}$, *dist* is the Euclidean distance between $x_{ij}$ and $u_j$. The anomalous HF selection method chooses those bands with a smaller *var$_j$* as the input data for anomaly detection.

iForest Building for Each Subclass and Data Streaming

Given the subclass and the sample to be detected, $I = \{Y_{sub\_j}, e\}$, where $e$ is the sample to be detected. Anomalous HFs are selected from $I$, represented as $I' = \{Y_{sub\_j}', e'\}$, to build an iForest and calculate an anomaly score [41]. Details of the construction of iForest and iTree can be found in Algorithms 2 and 3, respectively, where exNode{ } is either an external node with no child, and inNode{ } is an internal node with one test and exactly two daughter nodes (*Left, Right*) in Algorithm 3. The anomaly scores for the target to be detected are recorded as $S_j \in R^{each\_j+1}$, which is the average of the iForest score, where $j = [1, 2, 3, \ldots, n]$, and $n$ is the number of subclasses.

---

**Algorithm 2**: iForest (*I′*, *n_t*, *n_s*)

---

Inputs: *I′*—input data, *n_t*—number of *iTrees*, *n_s*—sampling size of *iTree*
Output: a set of n_t *iTrees*
1: Initialize *Forest*
2: for *i* = 1 to *n_t* do
3:          *sub-I′*     ←     sample (*I′*, *n_s*)
4:          *Forest*     ←     *Forest* U *iTree*(sub-I′, 0)
5: end for
6: return *Forest*

---

There are two input parameters in the iForest algorithm: the number of trees *n_t* and the sample size of iTree *n_s*. The *n_s* controls the training data size. In this paper, when the total number of observed HTs samples and data streaming is less than 500, *n_s* is the number of all samples; otherwise, *n_s* is 500. The *n_t* represents iForest size, which affects the training time and the detection accuracy of the algorithm and will be discussed in Section 4.

---

**Algorithm 3**: *iTree* (*I′*, *h*)

---

Inputs: *I′*—input data, *h*—current tree height

---

Output: an *iTree*
1: if | *I′* | ≤ 1 or *e′* is isolated then
2:       return exNode{Size     ←     |*I′*|}
3: else
4:       let *B* be the selected anomalous HFs
5:       randomly select a band *b* ∈ *B*
6:       randomly select a value *a* from *max* and *min* values of band *b* in *I′*
7:       *I′_l*     ←     *filter*(*I′*, *b* < *a*)
8:       *I′_r*     ←     *filter*(*I′*, *b* ≥ *a*)
9.       return inNode{*Left*     ←     iTree(*I′_l*, *h* + 1),
                              *Right*     ←     iTree(*I′_r*, *h* + 1),
                              *Split band*     ←     *b*,
                              *Split Value*     ←     *a*}
End if

---

New Class Detection

The new class detection result is determined by the anomaly score $\{S_1, S_2, \ldots, S_j, \ldots, S_n\}$ obtained based on the improved iForest algorithm, where $S_j$ is the anomaly score of the *j*-th subclass and the sample to be detected *e*. We denote the maximum and average anomaly scores in the subclass as $S_{max}$, $S_{mean}$. The anomaly score of the sample to be detected is $S_e$. When $S_e$ satisfies the relationship of Equation (4), it means that *e* is an anomaly target compared with the subclass.

$$S_e > S_{max}, \quad S_e > S_{mean} + \lambda(S_e - S_{max}) \tag{4}$$

If the *e* belongs to the anomaly target relative to all subclasses, the detected data are considered to be a new class. It can be seen from Equation (4) that $S_e$ should meet two conditions. Suppose the target to be detected is a new unobserved target type. In this case, the corresponding anomaly score $S_e$ of the detected target with abnormal characteristics should be higher than all the anomaly scores of the observed samples in the subclass and further higher than the maximum anomaly score $S_{max}$ of the subclass. The maximum value of the observed subclass has a particularity. To make the detection result more reliable, the $S_e$ of the new target to be detected should be greater than the average value $S_{mean}$ of the observed subclass. $\lambda(S_e - S_{max})$ defines the difference in anomaly scores between the $S_e$ and $S_{mean}$. When the value of $\lambda(S_e - S_{max})$ is larger, it means that the target's anomaly score is much higher than $S_{mean}$ can be detected as a new type. In this case, when Formula (4)

is satisfied, the high probability is a new type target, but the new unobserved type target with a low anomaly score will be wrongly judged as a known target. On the contrary, if the value of $\lambda(S_e - S_{max})$ is relatively small, the real new type samples can be detected with high probability. However, the observed targets with high abnormal scores that are not new classes will be wrongly judged as a new type. Therefore, the value of $\lambda(S_e - S_{max})$ should be between $S_e$ and $S_{mean}$—neither too large nor too small. We find that the value of $\lambda$ in the hypersonic data set is between 2/3 and 1/4, which can ensure the detection accuracy and low false alarm rate, so the value of $\lambda$ is 1/3 in this paper.

## 4. Experiment and Analysis

The experiments were conducted on simulated HFs of HTs to illustrate the performance of the proposed method.

### 4.1. Simulated Hyperspectral Data Sets of Hypersonic Targets

The hyperspectral radiation characteristics of HTs of different classes and flight states given in Section 2 were obtained from the theoretical model. Early in the development of hyperspectral detectors, researchers hypothesized that the HFs of the same targets are unique. However, as well as laboratory data, the variability in the radiation spectrum of most targets was observed. Many mechanisms may lead to the observed variability, including uncompensated errors in the sensor and uncompensated atmospheric and environmental effects. For the HFs of HTs, in addition to the influence of the above factors, the changes in the atmospheric parameters of the actual flight state will also reduce HF changes of the same type with a fixed flight state. Therefore, these combined factors lead to the observed HFs of HTs in the same class and flight state comprising a set of random curves that change according to a certain rule. This was proven in Ref [42]. The mean vectors and covariance matrices of the HFs of subclass $k$ are represented as $P_k$ and $\sum_k$. The eigenvalues and eigenvectors of covariance matrices $\sum_k$ are computed and arranged as diagonal matrices $\Lambda_k$ and column matrices $\Phi_k$, respectively. Thus, the $i$-th sample of the $k$-th subclass can be expressed as

$$P_i = \overline{P_k} + \Phi_k \Lambda_k^{1/2} R_i \tag{5}$$

where $R_i$ is the spatial correlation matrix, and $\Phi_k \Lambda_k^{1/2}$ reflects the intraclass variability of HFs. The larger the value, the more drastic the change of HFs within a class. According to the actual observation experience of space-based hyperspectral detectors, the value of $\Phi_k \Lambda_k^{1/2}$ was set from 0 to 0.2, randomly. $R_i$ is the spatial correlation matrix, whose value is a random number between 0 and 1. We simulated 400 observation samples for each flight state of RAM-C II and HTV-2, from which 5 samples were randomly selected, as shown in Figure 10. In the subsequent algorithm analysis, the samples of RAM-C II were set as a known observation class to verify whether the sample could be detected as a new class when the HTV-2 data stream sample appeared.

Comparing Figure 10I,II under the same flight state, we noted that the HFs of different classes were relatively similar. For example, the HF difference between RAM-C II of 50 km 25 Mach and HTV-2 of 50 km 25 Mach was much smaller than that of RAM-C II of 50 km 20 Mach, which further indicates that the intraclass variability of HFs is much larger than the interclass variability for HT. Each sub-figure in Figure 10 shows the difference of HFs from the same class of HT in the same flight state, which made it more difficult to detect whether an HT belonged to a new class or an observed known class.

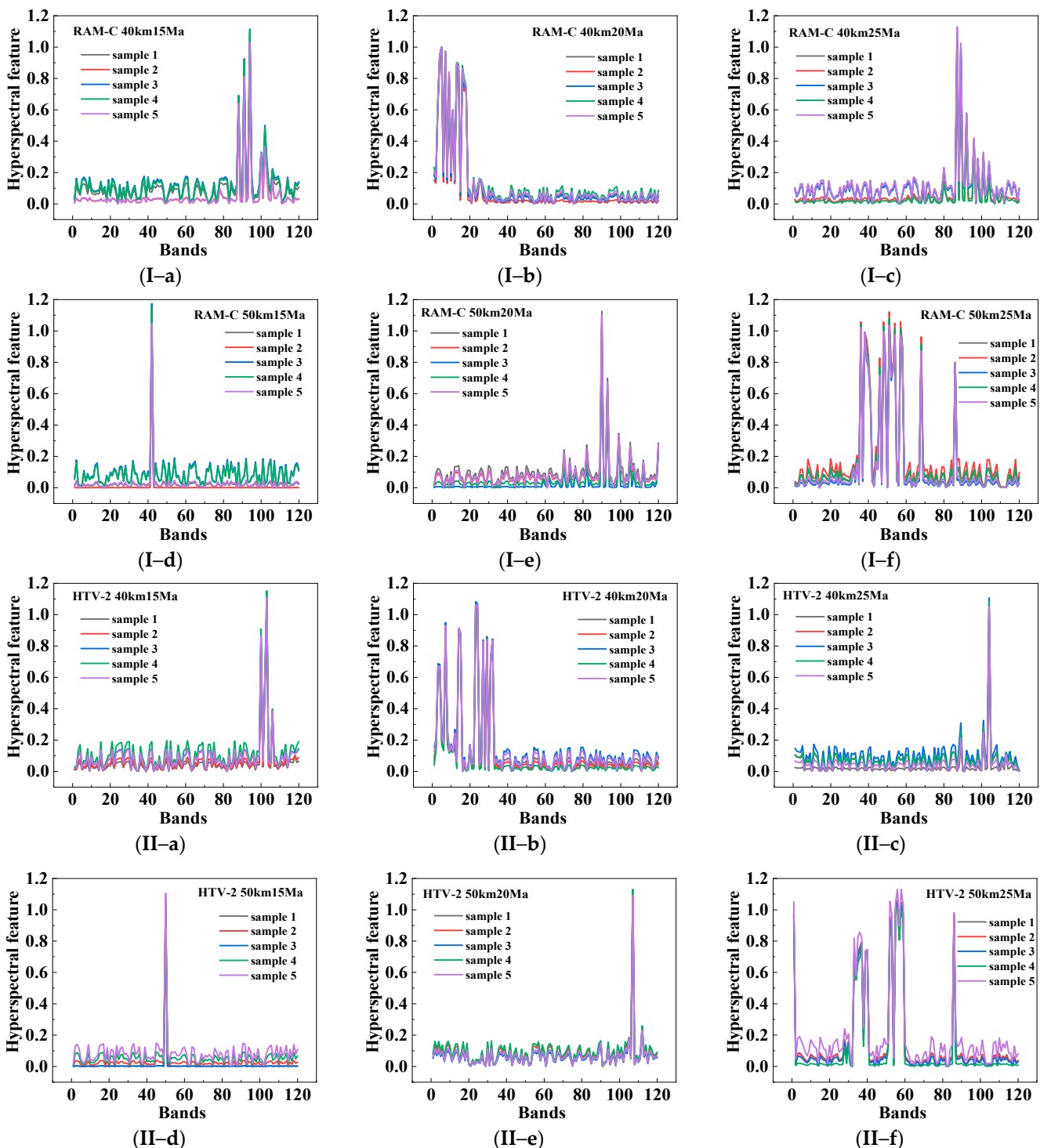

**Figure 10.** The simulated observation samples for each flight state of RAM-C II and HTV-2. (**I**) RAM-C II. (**II**) THV-2. (**a**) 40 km 15 Mach. (**b**) 40 km 20 Mach. (**c**) 40 km 25 Mach. (**d**) 50 km 15 Mach. (**e**) 50 km 20 Mach. (**f**) 50 km 25 Mach.

### 4.2. The Analysis of Detection Accuracy

The performance of the proposed new class detection algorithm was analyzed by taking the RAM-C II samples as the observed data. The simulation conditions are given, as shown in Table 2. For each subclass, the parameters of the iForest were as follows: the number of anomalous HFs was 6, which is generally an empirical value, and it was 5% of the entire number of spectral bands. The number of iForest was 10, the number of iTrees in each iForest was 4, and the node in each iTree was all samples.

**Table 2.** Experiment condition settings.

| Conditions | Observed Samples | | Data Streaming |
|:---:|:---|:---:|:---:|
| | **Flight States** | **Number** | **Class** |
| 1 | RAM-C II 40 km 15 Ma | 400 | HTV-2 |
| | RAM-C II 40 km 20 Ma | 400 | |
| | RAM-C II 40 km 25 Ma | 400 | |
| | RAM-C II 50 km 15 Ma | 400 | |
| | RAM-C II 50 km 20 Ma | 400 | |
| | RAM-C II 50 km 25 Ma | 400 | |
| 2 | RAM-C II 40 km 15 Ma | 100 | HTV-C |
| | RAM-C II 40 km 20 Ma | 400 | |
| | RAM-C II 40 km 25 Ma | 200 | |
| | RAM-C II 50 km 15 Ma | 400 | |
| | RAM-C II 50 km 20 Ma | 300 | |
| | RAM-C II 50 km 25 Ma | 400 | |
| 3 | RAM-C II 40 km 15 Ma | 400 | RAM-C |
| | RAM-C II 40 km 20 Ma | 400 | |
| | RAM-C II 40 km 25 Ma | 400 | |
| | RAM-C II 50 km 15 Ma | 400 | |
| | RAM-C II 50 km 20 Ma | 400 | |
| | RAM-C II 50 km 25 Ma | 400 | |

The performance of unsupervised classification based on the process in Section 3 is shown in this section. It can be seen from Table 2 that the observed samples of each typical flight state of RAM-C II in conditions 1 and 3 were consistent, while condition 2 was non-uniform. The results of the subclass centers of observed samples in conditions 1 and 3 are shown in Figure 11a, and the subclass centers in condition 2 are shown in Figure 11b. The red points in Figure 11 represent the subclass centers, and the remaining samples can be divided into subclass centers according to the distance between the center and samples. The proposed subclass definition approach was compared with the traditional FCM [43], DBSCAN [44], GMM [45] and HAC [46]. Since the FCM, GMM and AHC algorithms must manually give the number of clusters before classification, we set the number of cluster parameters to 6. DBSCAN can automatically determine the class number. Table 3 gives the average value of overall accuracy (OA), average accuracy (AA) and Kappa coefficient of four classification algorithms (over 20 runs), and the values of optimal classification results are marked in bold. It can be seen from Table 3 that the FCM, DBSCAN and GMM unsupervised classification algorithms were not suitable for HTs classification because of the intraclass variability of HFs of HTs. Although HAC achieved accurate classification under conditions 1 and 3, the classification accuracy decreased in condition 2 when the data distribution was not uniform. Only the proposal achieved the correct unsupervised classification of all samples in all conditions, which verified its robustness.

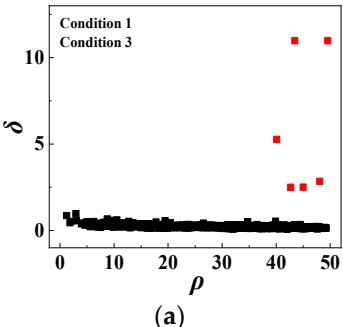
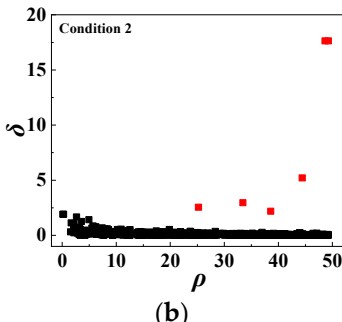

(a)          (b)

**Figure 11.** The result of subclass center after unsupervised classification. (**a**) Condition 1 and 3. (**b**) Condition 2.

**Table 3.** The average value of OA AA and Kappa coefficient.

| Data Sets | Evaluation Index | FCM | DBSCAN | GMM | HAC | Proposal |
|---|---|---|---|---|---|---|
| | OA | 74.6% | 100% | 50.0% | **100%** | **100%** |
| Condition 1, 3 | AA | 75.0% | 100% | 25.0% | **100%** | **100%** |
| | Kappa | 69.5% | 100% | 40.0% | **100%** | **100%** |
| | OA | 74.8% | 83.3% | 33.3% | 73.6% | **100%** |
| Condition 2 | AA | 75.0% | 75.0% | 22.2% | 75.0% | **100%** |
| | Kappa | 69.8% | 80.0% | 20.0% | 68.3% | **100%** |

The proposed USD-EiForest was performed on the data streaming and observed HTs under three conditions, as seen in Table 2. In this section, the improved anomaly detection algorithm was performed on the six subclasses obtained from Figure 11 and the data to be detected. The anomaly score of the HTs is shown in Figure 12, where the purple line represents the anomaly score of the observed HTs, the red line represents the anomaly score of the sample to be detected, and the yellow line represents the mean value of the score of the observed target. According to the detection criterion of Equation (4) in Section 3.2.2, the sample to be detected, belonging to the anomaly target compared with each subclass in condition 1 and condition 2, was a new class. For condition 3, it can be seen in Figure 12(3–a) that the anomaly score of the HT to be detected tended toward the mean value of the observed sample, which proved that the HFs of the detected target were close to the HF distribution of the accumulated HTs. Therefore, it belonged to the observed known class. Figure 12 shows the results of one experiment. We repeated the tests 100 times, and the detection accuracy of the proposal was 100%.

### 4.3. The Impact of Anomaly HFs Selection on the Proposal

The high mobility of the HTs required a high detection efficiency for DNHT. The detection efficiency was determined by the number of iForests $N_f$ and the number of isolated trees $N_t$ that comprised each isolated forest. Ref [22] points out that the method of iForest usually works well before $N_t = 100$, even if the $N_f$ is only 1 in practice. Therefore, given a new data set stream, the iForest can work as long as $N_t = 100$, $N_f = 1$. For the EiForest method proposed in this paper, the value of $N_t < 100$ can achieve anomaly target detection. However, the minimum value of $N_t$ that ensures the detection accuracy needs to be analyzed according to the specific data set. As long as $N_t$ ensures the accuracy of the proposed algorithm, $N_f$ can set any value. In this section, the node in each iTree was all samples. The detection accuracies of the proposal with two conditions were compared; anomalous hyperspectral features were selected (AHFS) or not selected (AHFNS) under changed $N_f$ and $N_t$, and the minimum values of $N_f$ and $N_t$ were given to accurately detect new classes in subclasses of RAM-C II under flight state 40 km 25 Mach.

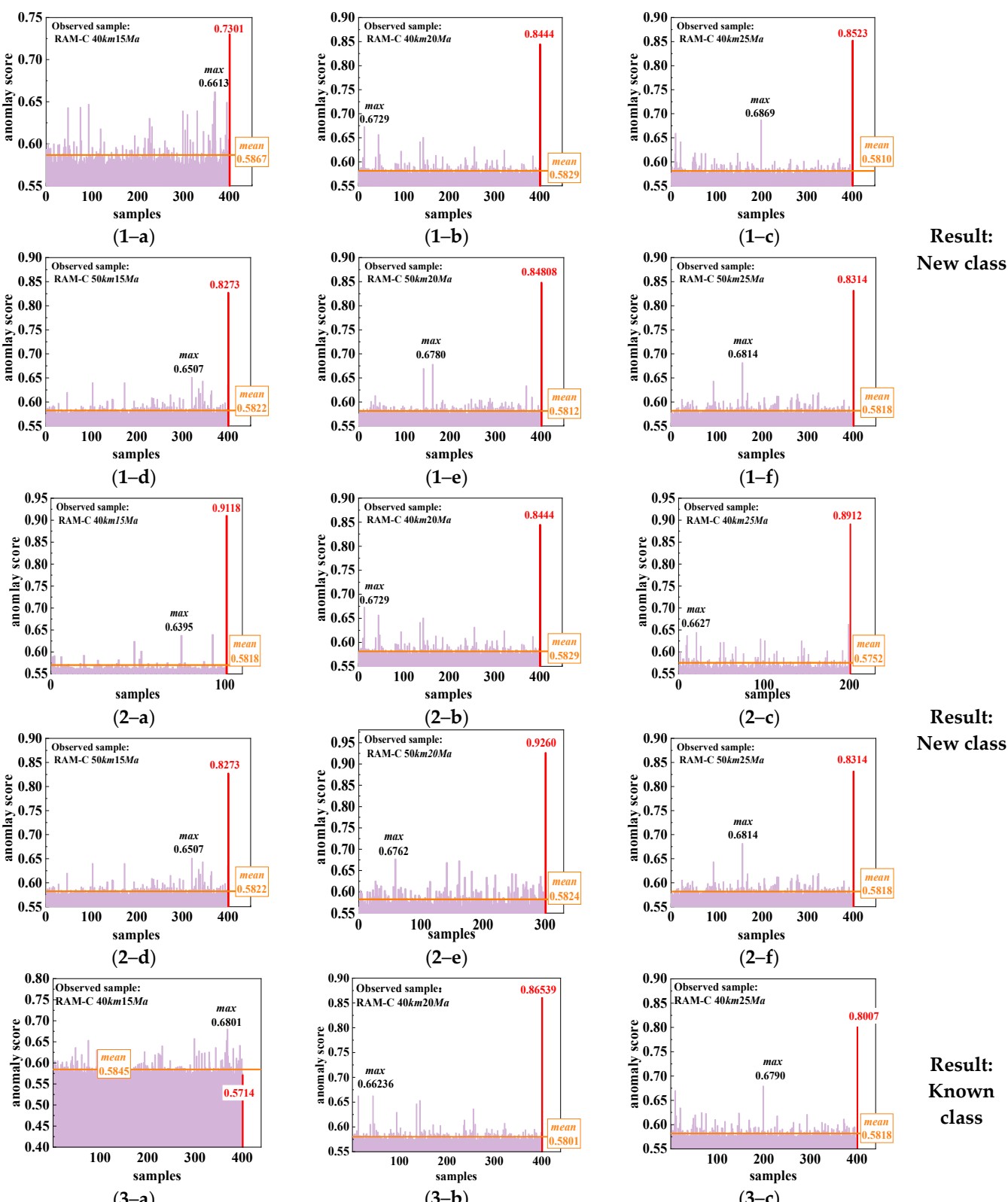

**Figure 12.** *Cont.*

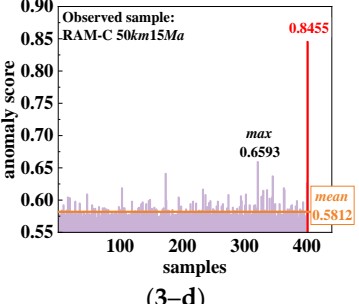 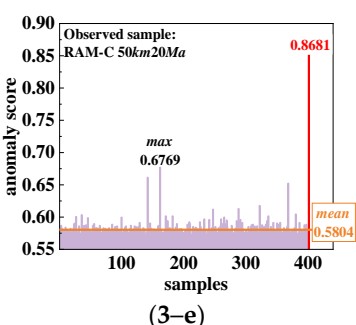 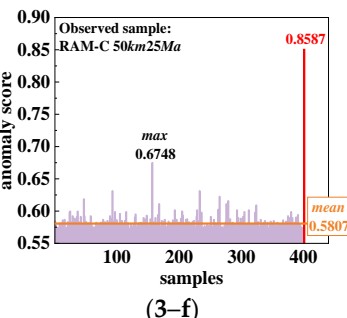

(**3–d**)        (**3–e**)        (**3–f**)

**Figure 12.** Anomaly scores of subclasses and data streaming. (**1**) Condition 1. (**2**) Condition 2. (**3**) Condition 3. (**a**) 40 km 15 Mach. (**b**) 40 km 20 Mach. (**c**) 40 km 25 Mach. (**d**) 50 km 15 Mach. (**e**) 50 km 20 Mach. (**f**) 50 km 25 Mach.

Figure 13 shows the anomaly score results of the subclass of RAM-C II under flight state 40 km 25 Mach and the HTV-2 sample to be detected, which were randomly selected from 100 run times of the experiment. The values of $S_{max}$ and $S_e$ directly determined the detection accuracy of the proposal. Therefore, the discussion of the result in Figure 13 mainly revolved around $S_{max}$ and $S_e$, and the following conclusions can be drawn from the result. (1) Except for condition a($N_f = 1$, $N_t = 1$), the anomaly score of HTV-2 $S_e$ was always higher than 0.9 after AHFS, while the $S_e$ was lower than 0.9 with AHFNS. This proved that the anomaly characteristic of the sample of HTV-2 was more obvious after AHFS. (2) When fixing each set of values of $N_f$ and $N_t$, it can be seen from Figure 13I,II that the values of $|S_e - S_{max}|$ after AHFS were always greater than the value of $|S_e - S_{max}|$ with AHFNS. The larger the difference between $S_e$ and $S_{max}$, the more beneficial it was for new class detection, according to Equation (4). (3) Under the condition of AHFNS, the value of $|S_e - S_{max}|$ changed slowly with the increase in $N_f$ or $N_t$. However, the difference between $N_f$ and $N_t$ could increase rapidly and tended to be stable under AHFS.

Figure 13 shows the effect of AHFS on detecting the anomaly target in the subclass, and Table 4 further reveals the impact of AHFS on the proposed new class target detection algorithm. Under $N_f > 1$ or $N_s > 1$, the detection accuracy was improved by up to 38–61% with AHFS. When $N_f = 1$ and $N_s = 1$, the improvement in detection accuracy was not obvious. Because the number of iForests and iTrees was too small, the anomaly score of HTV-2 was similar to the observed subclass samples even after AHFS.

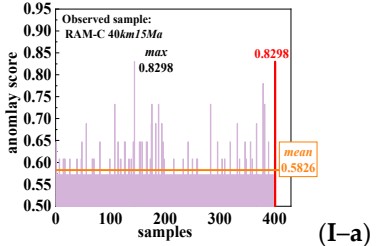 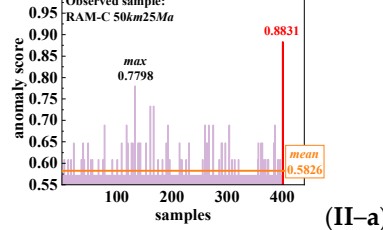

(**I–a**)                  (**II–a**)

**Figure 13.** *Cont.*

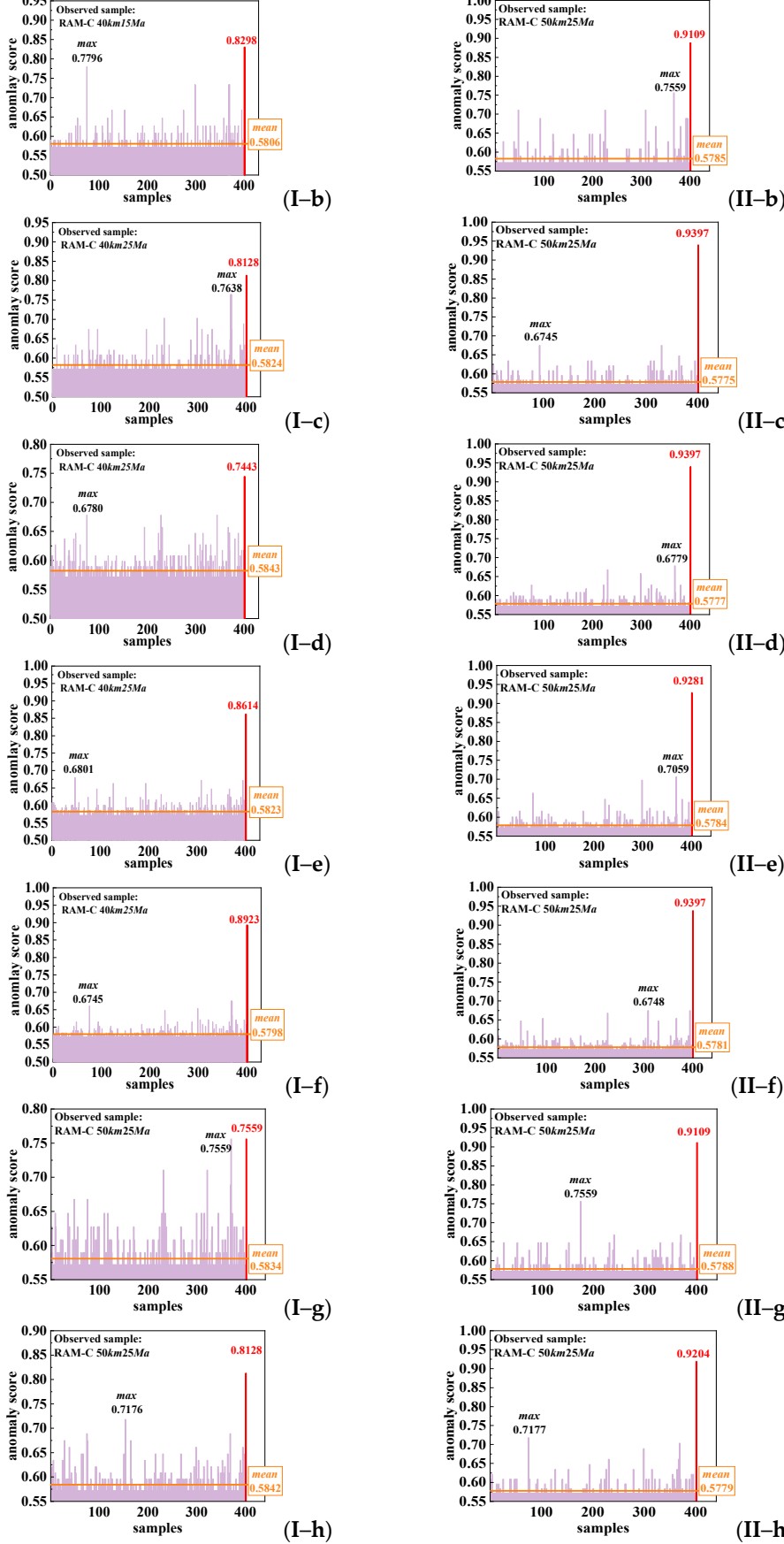

**Figure 13.** *Cont.*

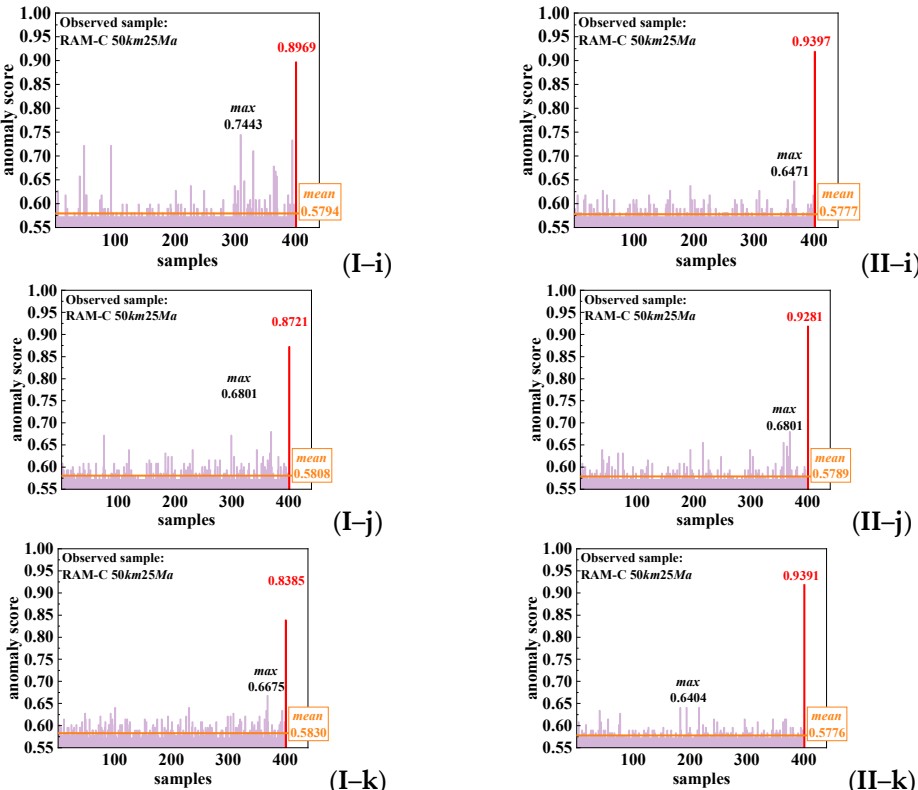

**Figure 13.** Anomaly scores of subclasses and data streaming under different value of $N_f$ and $N_t$. (**I**) AHFNS. (**II**) AHFS. (**a**) $N_f = 1$, $N_t = 1$. (**b**) $N_f = 2$, $N_t = 1$. (**c**) $N_f = 3$, $N_t = 1$. (**d**) $N_f = 4$, $N_t = 1$. (**e**) $N_f = 5$, $N_t = 1$. (**f**) $N_f = 6$, $N_t = 1$. (**g**) $N_f = 1$, $N_t = 2$. (**h**) $N_f = 1$, $N_t = 3$. (**i**) $N_f = 1$, $N_t = 4$. (**j**) $N_f = 1$, $N_t = 5$. (**k**) $N_f = 1$, $N_t = 6$.

**Table 4.** The detection accuracy of the proposal with different $N_f$, $N_t$ values.

| | **Detection Accuracy** | | |
| | **AHFS** | **AHFNS** | **Accuracy Improvement** |
|---|---|---|---|
| iForest = 1 iTree = 1 | 11% | 3% | 8% |
| iForest = 1 iTree = 2 | 76% | 15% | 61% |
| iForest = 1 iTree = 3 | 83% | 28% | 55% |
| iForest = 1 iTree = 4 | 90% | 31% | 59% |
| iForest = 1 iTree = 5 | 96% | 44% | 52% |
| iForest = 1 iTree = 6 | 97.7% | 54% | 43.7% |
| iForest = 2 iTree = 1 | 61% | 23% | 38% |
| iForest = 3 iTree = 1 | 77% | 29% | 48% |
| iForest = 4 iTree = 1 | 86% | 46% | 40% |
| iForest = 5 iTree = 1 | 97% | 48% | 49% |
| iForest = 6 iTree = 1 | 99% | 54% | 45% |

To determine the minimum values of $N_f$ and $N_t$ of the proposal, the values of $N_f$ and $N_t$ were increased gradually to observe the detection accuracy under conditions of AHFNS and AHFS. The gray curves are the contour lines, as shown in Figure 14, indicating the number of iTrees and iForests on the gray curves that corresponded to the detection accuracy of the proposed algorithm. When the values of $N_f$ and $N_t$ did not exceed 4 after AHFS, the detection accuracy approached 100%. For the condition of AHFNS, when the values of $N_f$ and $N_t$ were 20, the detection accuracy had not yet reached 100%. It can be concluded that the complexity of the new class detection algorithm can be greatly reduced, and the detection efficiency can be improved through AHFS, making the proposed algorithm more suitable for detecting the new class of HTs.

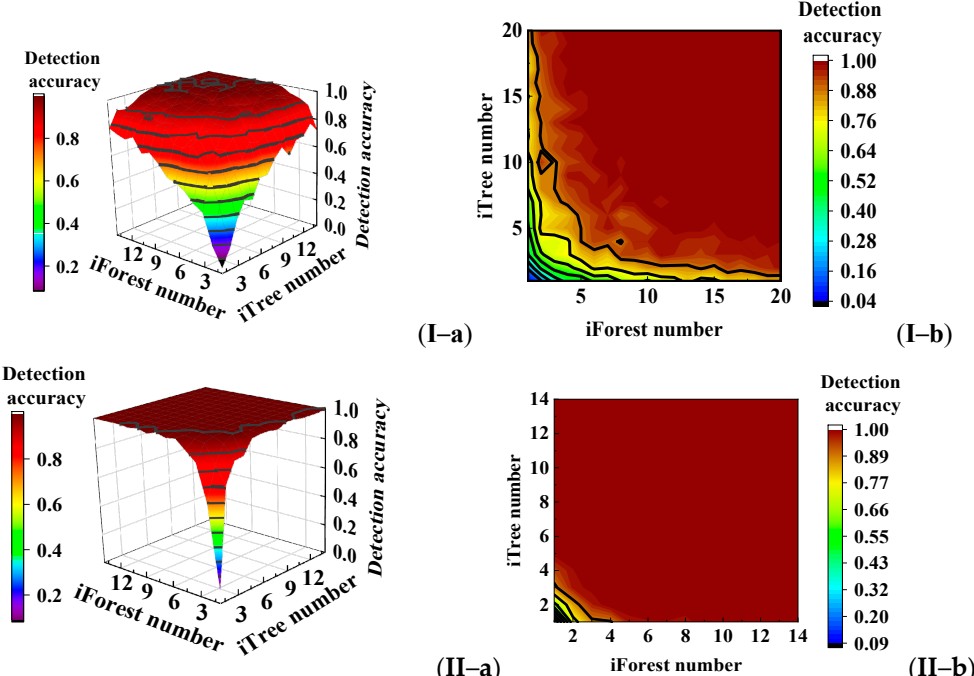

**Figure 14.** Detection accuracy of the proposal under different values of $N_f$ and $N_t$. (**I**) AHFNS. (**II**) AHFS. (**a**) Three-dimensional map. (**b**) Top view two-dimensional map.

## 5. Conclusions

This paper addressed the HTs' new class detection problem under challenging conditions without new class samples and data sets. For this purpose, a USD-EiForest algorithm based on unsupervised subclass definition and efficient iForest was presented. Benefiting from the local anomaly characteristics of the new class HT sample relative to the subclasses, the proposal determined whether the sample to be detected was a new class for each subclass to realize the DNHT. Experimental evaluation of the proposed algorithm on simulated HFs of RAM-C II and HTV-2 HTs illustrated that the proposal could detect a new class of HT with high detection accuracy and efficiency. The unsupervised subclass division algorithm divided the observed HT samples into multiple subclasses, and the $N_f$ and $N_t$ of EiForest were significantly reduced compared with the traditional iForest, improving the detection efficiency. However, the algorithm proposed in this paper can only determine whether a sample is a new class at present. With the construction of HFs data sets of HTs, it will be necessary to develop an algorithm that enables both new class detection and known class classification in the future.

Moreover, the research contribution of this paper lays the theoretical foundation for the new class detection of HTs based on HFs. Because of the limitation of the military value of hypersonic targets, we could only verify the algorithm on a simulation data set. However, the spectral radiation characteristics of the simulated data set were calculated based on the real measured flow field data, allowing the simulated hyperspectral data set and the real observed data set to maintain a high consistency. Therefore, with the development of hypersonic targets in the future, the proposed method in this paper may be applied to the actual scene with a suitable space-based detector. The research in this paper is based on the unlabeled observation data set to detect the new types of samples that have not been observed. The classification and labeling of observed samples, construction corresponding to the data set, and supporting type recognition of known hypersonic targets are the work that we carried out and will be published later.

**Author Contributions:** Conceptualization, L.S.; methodology, S.Y. and Y.Z.; software, S.Y.; validation, B.Y.; formal analysis, S.Y.; investigation, F.L.; resources, L.S. and Y.D.; data curation, S.Y.; writing—original draft preparation, S.Y.; writing—review and editing, L.S.; visualization, Y.Z.; supervision, Y.D.; project administration, L.S.; funding acquisition, L.S. All authors have read and agreed to the published version of the manuscript.

**Funding:** This research was funded by the National Natural Science Foundation of China (Nos. 61871302, 62101406, and 62001340), the Innovation Capability Support Program of Shaanxi (Program No. 2022TD-37) and the Fundamental Research Funds for the Central Universities (No. JB211311).

**Data Availability Statement:** Not applicable.

**Acknowledgments:** The authors would like to thank the handling editor and the anonymous reviewers for their careful reading and helpful remarks.

**Conflicts of Interest:** The authors declare no conflict of interest.

## Abbreviations

| Original text | Acronyms and abbreviations |
| --- | --- |
| Hypersonic target | HT |
| Unsupervised subclass definition and efficient isolation forest based on anomalous hyperspectral feature selection | USD-EiForest |
| Hyperspectral feature | HF |
| Detection of the new class of hypersonic targets | DNHT |
| Density peak clustering | DPC |
| Line of sight | LOS |
| Low earth orbit | LEO |
| Principal component analysis | PCA |
| Improved efficient iForest | EiForest |
| Fast Fourier transform | FFT |
| Isolation forest | iForest |
| Isolation trees | iTree |
| Overall accuracy | OA |
| Average accuracy | AA |
| Anomalous hyperspectral features are selected | AHFS |
| Anomalous hyperspectral features are not selected | AHFNS |

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
