# Peer review of "Detection of the New Class of Hypersonic Targets under Emerging Hyperspectral Sample Streams: An Unsupervised Isolation Forest Solution"

_remotesensing, doi:10.3390/rs14205191_

Round 1

Reviewer 1 Report (Previous Reviewer 1)

Dear authors,

Thank you very much for sending me correct and well-argued answers in terms of theory as well as practical applicability to all six of my comments and suggestions.

Author Response

Dear Editor-in-Chief, Editors and Reviewers,

On behalf of my co-authors, we thank you very much for giving us a precious opportunity to revise our manuscript, we really appreciate editors and reviewers from the bottom of heart for their positive and constructive comments, as well as for their invaluable and careful suggestions on our manuscript entitled "Detection of the new Class of Hypersonic Targets Under Emerging Hyperspectral Sample Streams: An unsupervised isolation Forest Solution" (remotesensing-1939615).

We have studied reviewers’ comments under scrutiny, upon which we have made due modifications as well as further detailed explanations and corrections. The modifications are marked in RED of our response, and corresponding responses to each comment are highlighted in BLUE for easy reviewing.

We are uploading (a) our item-by-item response to the comments (below) (response to reviewers), (b) an updated manuscript with yellow highlighting indicating changes, and (c) a clean updated manuscript without highlights (PDF main document). 

Again, we would like to express our great appreciation to the Editor-in-Chief as well as the reviewers and editors for those wise suggestions and dedicated hard works far beyond the call of duty, we shall always remain grateful! We hope our responses and the revised manuscript meet with your positive approval. Kindly favor us with a reply at your earliest convenience.

Thank you and best regards.

Shurong Yuan, Lei Shi, Bo Yao, Yutong Zhai, Fangyan Li, Yuefan Du

Reviewer 2 Report (Previous Reviewer 3)

The content of the revised manuscript allows for the illustration of a proof of concept (POC) on experimental (synthetically generated) data.

The whole is better introduced and described in more detail than in the previous version. As a result, it becomes more accessible and ultimately readable.

However, the proof of concept remains limited to the only detection of a new class. Indeed, the algorithm currently suggested and described can only determine whether a new incoming sample represents a new class or not. This currently limits the scope of the proposed method since the classification on known classes is not taken into account.

Although the content is a bit weak (for above-mentioned reasons), the case study and the application field are fully innovative.

1) One critical question remains : given a new data set stream ( not seen yet), how to choose an appropriate value of N_f and N_t in practice remains an open question? Are the results presented here (with minimal values) general enough to generalize to any situation ?

2) Explain better the reasons and grounds behind equation (4) other than the numerical results of experiments presented in 4.C. This needs to be investigated more thoroughly.

3) Formula (1) is unreadable

4) In Algorithm 2: for the sake of completeness, define exNode{} and inNode{}

Typo(s):

random select an band

The impact of anomalou HFs selection on proposal

Author Response

    Thank you very much for your enlightening suggestions on the revision of our manuscript, which are very comprehensive and targeted. We have made due modifications and corresponding explanations following your comments in word, and we feel that the logic and integrity of the revised manuscript are clearer.

    Again, we would like to express our great appreciation to the reviewers for those wise suggestions and dedicated hard work far beyond the call of duty, we shall always remain grateful! We hope our responses and the revised manuscript meet with your positive approval. 

Round 2

Reviewer 2 Report (Previous Reviewer 3)

The Authors have corrected their manuscript according to my requests and expectations. They were able to add the responses which allow to better appreciate and highlight their contribution and the significance of the work done. The content of the revised manuscript has thus been improved and strengthened. It now appears to me acceptable for publication.

This manuscript is a resubmission of an earlier submission. The following is a list of the peer review reports and author responses from that submission.

Round 1

Reviewer 1 Report

Dear authors,

I appreciate very much your great efforts involved in this research work. I found this paper interesting for its valuable scientific content, both theoretically and practically. Also, it has excellent potential for visibility since it can captivate the attention of a large community of readers and specialists working in the field. However, there is plenty of room to improve your manuscript's overall quality. Perhaps the following comments and suggestions can help achieve this:

1. To highlight the accuracy of the proposed detection algorithm, please compare the accuracy of the results obtained in your research work through a table inserted in the manuscript with the results obtained by different methods reported in the specialized literature.

2. For a better understanding by the reader of the manuscript's content, it is necessary to insert a table with the acronyms and abbreviations used in the text.

3. Please explain briefly the impact of the parameters of the gas model on the accuracy of both algorithms proposed for detection. 

4. Please give some details on the transmission model established based on MODTRAN software to reflect the atmospheric attenuation effect of spectral transmission.

5. A rigorous robustness analysis of the accuracy and efficiency of the proposed  USD-EiForest detection algorithm based on unsupervised subclasses definition and efficient iForest to changes in the values of the main model parameters with a significant impact could be an excellent challenge for increasing the overall quality of the manuscript. 

6. For the readers and specialists in the field interested in algorithm implementation, some details on the software and its version used for simulations are helpful. 

Thanks, 

Reviewer 2 Report

General comments

I have been invited to review this manuscript by Yuan et al. The manuscript aimed to address hypersonic targets new class detection problems in challenging conditions without new class samples and data sets. Despite not being particularly a specialist in the field of using remote sensing for military applications, I think that the paper suffers in many aspects. The manuscript’s structure needs a complete revision since there is no clear aim in the introduction, no clear method section, no discussion section and extra huge figures. Furthermore, I appreciate the authors are not probably native English speakers, but the paper needs a major round of English revision. As it stands, the paper is not interesting to read and follow. Please see my specific comments below:

Title

Lines 2-4: The first letters of some words are capitalised (e.g., Forest), whilst they are small in other words (e.g., isolation).

Abstract

Line 11: The first letters should be capitalised in “hypersonic targets”.

Line 13: Why “Forest” is capitalised?

Line 16: The first letters should be capitalised in “hyperspectral features”.

Lines 16-18: “Specifically, the phenomenon is first revealed that the hyperspectral features (HFs) of new class HTs have no anomaly characteristic compared to the globally observed samples and have prominent anomaly characteristics relative to the local subclasses of observed samples”.  This statement is very complicated to understand and perhaps needs English revision.

Line 18: “Second”? What is “first” then?

Line 22: “DNHT” Capitalise the first letters.

Line 23: What are “RAM-C II” and “HTV-2”?

Keywords

Lines 26-27: Most, if not all your keywords exist in the title. This is likely because it is a lengthy title. Consider choosing different keywords as much as possible.

Introduction

Line 32: “HTs” should be moved after “Hypersonic targets” in line 30.

Lines 30-32: Ref(s) needed.

Line 41: “HFs” Spell out when first mentioned in a section.

Lines 41-48: Ref(s) needed.

Line 47: “DNHT” Spell out when first mentioned in a section.

Lines 57-60: The Refs numbers are missing.

Line 65: “open-set recognition” Capitalise the first letters.

Line 67: Ref no. 24 comes before Ref no. 23!

Line 100: “section 4”.

Main comment: It is unclear what is the main purpose of the study. This should be clearly mentioned in the last paragraph of the introduction with what gaps you are trying to fill in the literature.

Methods

Line 102: Is this your “materials and methods” section?

Results

Line 337: Figure 10: What a huge figure to follow up!

Line 363: Figure 11: What a huge figure to follow up!

Reviewer 3 Report

I think that the content of this manuscript is interesting and that the problematic addressed is original and of current interest, but as it stands the actual writing is really very poor and lacks any effort at vulgarization and therefore clarity.

1) It is quite difficult to read. The content is too dense in places, lacks illustrations and leads to an overall impression of confusion, which is detrimental to the manuscript.

2) The Authors should definitely rework the content of their manuscript in depth to make it more accessible to readers not familiar with this specific field.

3) It is necessary to add sufficiently explicit diagrams to support the underlying ideas to improve the readability of the proposed approach.

4) There are also many figures presented, but ultimately very little detailed analysis of the content of these figures to support an objective evaluation of the proposed method.

5) The consequence of these shortcomings is that it is difficult to identify the contribution made by the authors, especially as there is no objective comparison provided.

6) In fact, the current content seems to be limited to a proof of concept on an homemade dataset and it is not clear how this could be exploited by others for reproduction or comparison with future works.

7) The authors should list and discuss the prior knowledge required for the implementation of their method, the parameters that are essential and how to set them appropriately

8) On the basis of the current content, it is also difficult to assess the strengths and weaknesses of the proposed approach and its limits of effectiveness. This aspect also needs to be strengthened